# Molecular Characterization of Associated Pathogens in Febrile Patients during Inter-Epidemic Periods of Urban Arboviral Diseases in Tapachula Southern Mexico

**DOI:** 10.3390/pathogens10111450

**Published:** 2021-11-08

**Authors:** Geovana Calvo-Anguiano, José de Jesús Lugo-Trampe, Gustavo Ponce-García, Angel Lugo-Trampe, Laura Elia Martinez-Garza, Marisol Ibarra-Ramirez, Luis Daniel Campos-Acevedo, Sandra Caballero-Sosa, Alan Esteban Juache-Villagrana, Ildefonso Fernández-Salas, Adriana Elizabeth Flores-Suarez, Iram Pablo Rodriguez-Sanchez, Karina del Carmen Trujillo-Murillo

**Affiliations:** 1Departamento Genética, Facultad de Medicina, Universidad Autónoma de Nuevo Leon, Av. Francisco I. Madero S/N, Mitras Centro, Monterrey 64460, Nuevo Leon, Mexico; gcalvo.ang@gmail.com (G.C.-A.); lugotramjose@hotmail.com (J.d.J.L.-T.); laelmar@yahoo.com.mx (L.E.M.-G.); m.ibarrar25@gmail.com (M.I.-R.); luisdanielc@yahoo.com (L.D.C.-A.); 2Facultad de Ciencias Biologicas, Universidad Autonoma de Nuevo Leon, Av. Pedro de Alba S/N, Ciudad Universitaria, San Nicolas de los Garza 66455, Nuevo Leon, Mexico; gponcealfa@gmail.com (G.P.-G.); alan.juache@gmail.com (A.E.J.-V.); ildefonso.fernandezsl@uanl.edu.mx (I.F.-S.); adrflores@gmail.com (A.E.F.-S.); iramrodriguez@gmail.com (I.P.R.-S.); 3Facultad de Medicina Humana “Dr. Manuel Velasco Suárez”, Campus IV, Universidad Autónoma de Chiapas, Tapachula 30792, Chiapas, Mexico; angel.lugo@unach.mx; 4Clínica Hospital Dr. Roberto Nettel Flores, Instituto de Seguridad y Servicios Sociales de los Trabajadores del Estado, Av. Tuxtepec y Oaxaca S/N, Francisco Villa, Tapachula 30740, Chiapas, Mexico; sandyluzcs@hotmail.com; 5Centro de Investigación y Desarrollo en Ciencias de la Salud, Universidad Autónoma de Nuevo León, Av. Mutualismo, Monterrey 64460, Nuevo León, Mexico

**Keywords:** Arbovirus, Chikungunya, ZIKA, Leptospira, Flavivirus, Alphavirus, Mexico

## Abstract

Emerging and re-emerging vector-borne infections are a global public health threat. In endemic regions, fever is the main reason for medical attention, and the etiological agent of such fever is not usually identified. In this study, non-specific febrile pathogens were molecularly characterized in serum samples from 253 patients suspected of arbovirus infection. The samples were collected in the southern border region of Mexico from April to June 2015, and February to March 2016. ZIKV, CHIKV, DENV, leptospirosis, and rickettsiosis were detected by qPCR and nested PCR to identify flavivirus and alphavirus genera. The results indicated that 71.93% of the samples were positive for CHIKV, 0.79% for ZIKV, and 0.39% for DENV, with the number positive for CHIKV increasing to 76.67% and those positive for ZIKV increasing to 15.41% under the nested PCR technique. *Leptospira Kmetyi* was identified for the first time in Mexico, with a prevalence of 3.16%. This is the first report of ZIKV in Mexico, as well the first detection of the virus in early 2015. In conclusion, the etiological agent of fever was determined in 94% of the analyzed samples.

## 1. Introduction

Arbovirus is manifested as a wide variety of clinical symptoms in the host that can range from asymptomatic or oligosymptomatic infection to even death [1,2]. During the acute phases of arboviruses such as dengue fever (DENV), chikungunya (CHIKV), and zika (ZIKV), the clinical manifestations are similar [3,4,5,6,7]. This fact complicates correct identification. There are also reports of coinfection, which, coupled with temporal or geographical clinical variations, makes differential diagnosis even more challenging [1].

In addition to the complexity of clinical diagnosis, arboviruses coexist with several pathogens that cause febrile diseases, such as leptospirosis and rickettsiosis, which are also regionally endemic [8,9,10]. Therefore, it is essential to understand the origins of febrile reactions to identify and determine the level of transmission of each disease and to correlate different clinical profiles to estimate the true magnitude of the problem and thus provide specific information for epidemiological surveillance, as well as establish timely specific patient management.

Since disease patterns and the spread of infections change over time, it is important to increase access to relevant epidemiological information, provide alternative diagnostic methods, and expand the evidence needed for decisions related to preventive and diagnostic measures. In this study, pathogens that cause febrile symptoms were molecularly characterized in serum samples from patients with symptoms (such as chikungunya disease in the southern border city of Tapachula, Chiapas, Mexico).

## 2. Results

We analyzed 253 samples from patients with the suggestive clinical presentation of arbovirus infection from the Dr. Roberto Nettel Hospital. Two hundred and six samples were obtained in April–June 2015, and 47 in February–March 2016, in several locations near Tapachula, Chiapas State, Mexico. Detailed patient clinical and demographic data are provided in Figure 1, Appendix A. In total, 169 (66.8%) were female and 84 (32.2%) were male, with ages ranging from 7 to 74 years and a mean age of 39.4 years. Of the total samples, 72 (28.45%) corresponded to the 41–50-year-old age group. Distribution by sex was unequal, showing a male/female ratio of 0.49. We determined that 76.67% (n = 194) of sera were positive for CHIKV, and 15.41% (n = 39) were positive for ZIKV, highlighting the presence of coinfections in 1.98% (n = 5) of CHIKV/ZIKV samples. Other pathogens analyzed presented a prevalence of 3.16% (n = 8) for Leptospira, 0.40% (n = 1) for DENV, and 6.32% (n = 16) negative results. In the first period (April–June 2015), 12 patients with ZIKV were detected (three patients were detected in April). Moreover, eight patients with leptospirosis were detected.

The clinical descriptions of patients infected with ZIKV and CHIKV are shown in Figure 2 and Appendix A. The mean number of days between the onset of symptoms and consultation was 3 to 4, with no difference between ZIKV+ and CHIKV+. However, most of the patients (6/8) with leptospirosis experienced 6 or more days before symptom onset. Symptoms such as fever and headache were reported most frequently (>90%); however, there was no significant difference between the groups (*p* > 0.05). The symptoms that did show substantial differences between ZIKV+ and CHIKV+ patients included myalgia (65.7% vs. 11.6%, *p* < 0.01), mild–moderate arthralgia (71.4% vs. 40.2%, *p* < 0.01), severe polyarthralgia (28.6% vs. 59.3%, *p* < 0.01), arthritis (17.1% vs. 54.5%, *p* < 0.01), retro-orbital pain (51.4% vs. 15.9%, *p* < 0.01), conjunctivitis (71.4% vs. 10.6%, *p* < 0.01), cough (42.9% vs. 18%, *p* < 0.01), taste alteration (42.9% vs. 79.4%, *p* < 0.01), rash (60% vs. 79.9%, *p* < 0.05), adenomegaly (25.7% vs. 45%, *p* < 0.05), and edema (14.3% vs. 37%, *p* < 0.05).

Symptoms such as vomiting, photophobia, and petechiae were less frequent. Only 28.5% (10/35) of ZIKV-positive patients developed symptoms that met the standard case definitions of ZIKV disease. In patients with CHIKV, 16.9% (32/189) presented symptoms that met the criteria for Chikungunya disease. For Group II, out of 47 patients, we recorded five CHIKV+/ZIKV + coinfection cases. 

On the other hand, patients detected to have leptospirosis presented more frequent symptoms such as fever, headache, mild–moderate arthralgia, pruritus, nausea, chills, and conjunctivitis.

## 3. Discussion

Between the years 2014 and 2016, ZIKV and CHIKV caused outbreaks throughout the South American continent. Since then, these viruses have been classified as a public health problem [11]. An important factor associated with these diseases is the presence and abundance of vector mosquitos; numerous Aedes aegypti populations are found in the region, and vector incrimination was demonstrated after the isolation of CHIKV and ZIKV from field mosquito collections [6,12,13]. The vertical transmission of DENV was also documented by Danis-Lozano et al. (2019) in an extensive egg survey from south Mexico states [14]. Equally relevant is leptospirosis, a neglected tropical disease with signs and symptoms similar to those of arboviruses. Leptospirosis is included among the 17 neglected tropical diseases categorized by the World Health Organization [15]. 

Various reports on CHIKV, within the same period and region [3,4], showed a low detection rate of 55.3% and 64%, respectively, when using molecular methods. In this study, the detection rate was 88.83% (183/206). The same effect was observed with ZIKV, for which we reported 57.44% (27/47) prevalence, while Guerbois et al. [6] reported a percentage of 21%. This difference may be because two molecular techniques (qPCR and nested PCR) were used in our study to increase diagnostic efficiency. However, although nested PCR is one of the most sensitive PCR techniques, it has a higher probability of false positives under two amplification cycles. Therefore, care must be taken to reduce potential cross- and carryover contamination, which are the main difficulties associated with this technique [16].

One of the details observed in our work when using the probe for ZIKV in qPCR was low specificity because, in a subsequent analysis, a discrepancy was found in the sequence used (Appendix A). Although RT-qPCR probes are sensitive and specific, they often have delimiting factors. Previous studies have shown that sample processing and RNA extraction methods contribute significantly to false positive and negative results [17,18]. Similarly, some studies showed that sequence variation in viruses can affect detection by RT-qPCR [19]. According to the data observed in this study, in times of the emergence or reemergence of a pathogen, it is likely that qPCR assays will have specificity problems that must be addressed before such assays can be used as a reliable screening method. 

In our study, three people infected with ZIKV were detected around the middle of April 2015, while the first three cases of ZIKV in Mexico were reported on 25 November 2015 [20]. The virus was already detected in mosquitoes in the states of San Luis Potosi and Tabasco at the beginning of the same year [21]. Similarly, molecular clock studies showed that the virus was already circulating undetected as early as 2014 in Mexico and Central America [22,23]. Therefore, our data provide the first evidence for the existence of this virus in Mexico following the second quarter of the year.

Another objective of our study was to broaden the search for other pathogens that present symptoms similar to arboviruses [9]. Surprisingly, we detected eight cases of leptospirosis in 24 samples that were negative for flaviviruses and alphavirus. This finding is notable because leptospirosis is a globally distributed disease with a much higher risk in rural areas and urban slums in tropical areas. Rodents, dogs, cattle, pigs, and other wild animals are vectors of this disease [24,25]. In this study, the few diagnosed patients did not show any specific clinical characteristics of the disease, supporting the disease’s clinical similarities to other febrile illnesses [10]. In Mexico, studies on this disease are usually scarce, and the vast majority are retrospective [26,27,28,29], which highlights the paucity of data on the surveillance and epidemiology of leptospirosis needed to address this neglected disease.

The species detected in our study was *L. kmetyi,* with 87.38% identity, a finding that has already been reported by some studies [30,31,32] (Appendix A). However, most reported cases of leptospirosis are due to *L. interrogans*, *L. kirschneri*, and *L. noguchii* [33]. In some studies, the pathogenicity of *L. kmetyi*, which lacks most pathogenic-specific proteins and is not highly virulent in humans, was questioned [34]. Other studies classified *L. kmetyi* as a cause of disease in humans since *L. kmetyi* was isolated from symptomatic patients [30,35,36]. This study demonstrates, for the first time, the presence of *L. kmetyi* in patients from the border region of Mexico. This information provides important data for further investigations. More studies are needed to determine the serological and molecular characteristics of this strain and its distribution in Mexico, as well as explore the relationship between *L. kmetyi* and its possible role in human infection. We hope that this information will help create awareness about leptospirosis as an important emerging, neglected, tropical disease.

Regarding the standard clinical characteristics reported for CHIKV [37] and ZIKV [38], our data were very similar for CHIKV, with fever, arthralgia, chills, headache, rash, pruritus, nausea, arthritis, abdominal pain, diarrhea, pharyngitis, and edema being the main symptoms, as also reported by previous studies [3,4]. Taste disturbances, pharyngitis, and other symptoms were rarely described. For ZIKV, fever, headache, chills, arthralgia, conjunctivitis, nausea, myalgia, pruritus, rash, retro-orbital pain, and abdominal pain have been previously reported by other working groups [5,6,39,40].

As mentioned before, it is known that these two infections have similar clinical presentations; however, we detected some differences in their clinical manifestations. For ZIKV, symptoms included myalgia, mild–moderate arthralgia, retro-orbital pain, conjunctivitis, and cough. For CHIKV, severe polyarthralgia, arthritis, and abnormal taste were significantly different clinical symptoms between the two diseases. Despite these findings, when evaluating the clinical characteristics simultaneously, there is a considerable overlap of symptoms. This finding concludes that these signs or symptoms in isolation or as a group can increase the complexity of specific viral diagnosis.

Finally, despite the use of the different diagnostic techniques described above, 15 cases were negative for the aforementioned pathogens, which may have been due to a low viral load/bacterial load. This opens a path to search for other endemic pathogens in the region and develop panels for better identification [10,41]. Furthermore, although RT-qPCR is usually sufficiently specific and sensitive for virus identification, it is preferable to evaluate and optimize the assay with updated strains of the region being studied to reduce false-negative cases [42].

## 4. Materials and Methods

### 4.1. Biological Material

Serum samples were obtained from 253 patients from the Dr. Roberto Nettel Flores ISSSTE General Hospital in the city of Tapachula, Chiapas. The samples came from patients with febrile symptoms suspected of arbovirosis. After obtaining the sera, the samples were stored in 1.5 mL microtubes at a temperature of −80 °C, corresponding to two periods: April–June 2015, and February–March 2016.

### 4.2. Nucleic Acid Extraction

RNA was extracted from 140 μL of serum with a QIAamp Viral RNA Mini kit from Qiagen (cat #52906). DNA was extracted from 50 μL of serum using a Qiagen Gentra^®^ Puregene kit (#158445). The different extractions were carried out according to the manufacturer’s instructions. The extracted nucleic acids (RNA and DNA) were stored at −80 °C until use.

### 4.3. Arbovirus Detection Performed by One-Step RT-qPCR

Specific fluorescent assays were used to detect the presence of CHIKV, ZIKV, and DENV1-4. The primers/probes used for the detection of emerging arboviruses are described in Table 1. Final concentrations of the primers and probes for CHIKV, ZIKV, and DENV1-4 were set at 20×. RT-qPCR assays were carried out using a StepOnePlus device (Thermo Fisher Scientific, Waltham, MA, USA). A QuantiTect Probe RT-qPCR 1× (Qiagen, Valencia, CA, USA) was used with 1 μL of a 20× mix of primers/probes and 5 μL of RNA at a concentration of 30–500 ng in a volume of 20 μL under the following conditions: 30 min at 50 °C, 15 min at 95 °C, 45 cycles of 15 sec at 94 °C, and 60 sec at 60 °C. Data collected from the RT-qPCR assays were analyzed using system software. Samples that reached cycle threshold values (Ct) below 40 were established as positive.

### 4.4. Nested-PCR

Samples negative for CHIKV, DENV1-4, and ZIKV according to the RT-PCR assays were subsequently subjected to nested-PCR to identify flaviviruses and alphaviruses with universal primers. First, the negative samples were converted to cDNA using a high-capacity RNA-to-cDNA kit (# 4368814) according to the manufacturer’s instructions. A set of previously described primers was used (Scaramozzino et al. 2001) for the nested-PCR flavivirus, which produced a 250 bp amplicon in the first round and a 220 bp amplicon in the second round (Table 1). For the first PCR, cFD2-MAMD primers were used with a GoTaq G2 Hot Start Polymerase reagent (#M7405) at a final concentration of 3 mM MgCl_2_; for the second PCR, the primers cFD2-FS778 were used at a final concentration of 2 mM MgCl_2_. The reaction conditions were as follows: 2 min at 95 °C, 25 cycles of 10 s at 95 °C, 15 s at 54 °C, and 40 s at 72 °C (30 cycles for the second PCR). In both PCRs, 1 U GoTaq G2 Hot Start Polymerase and primers at 2 μM were used.

For the alphavirus nested PCR, a set of previously described primers (Grywna et al., 2010) was used, which produced a 210 bp amplicon in the first round and a 197 bp amplicon in the second round (Table 1). For the first PCR, the primers ALPHA-1-Fod/ALPHA-1-Rev were used with the GoTaq G2 Hot Start Polymerase reagent (#M7405) at a final concentration of 3 mM MgCl_2_; for the second PCR, the primers ALPHA- 2-Fod/ALPHA-2-Rev were used at a final concentration of 2 mM MgCl_2_. The reaction conditions for the first PCR were as follows: 5 min at 95 °C and 40 cycles of 10 s at 95 °C, 30 s at 55 °C, and 30 s at 72 °C. The second PCR conditions were 5 min at 95 °C and 35 cycles of 10 s at 95 °C, 30 s at 60 °C, and 30 s at 72 °C. In both PCRs, 1 U GoTaq G2 Hot Start Polymerase and primers at 2 μM were used.

### 4.5. Identification of Leptospirosis and Rickettsiosis

Negative samples in the previous tests were subjected to the identification of leptospirosis and rickettsiosis via qPCR with SYBR green, which used primers designed by Garcia-Ruiz et al., 2016 (Table 1). Maxima SYBR Green/ROX qPCR Master Mix was used at 1X (#K0221) along with 50 uM primers and 5 µL of DNA in a volume of 25 µL. The reaction conditions for the first PCR were 2 min at 50 °C, 10 min at 95 °C, and 50 cycles of 15 s at 95 °C with 60 s at 60 °C. 

### 4.6. Sequencing and BLAST

PCR products from nested-PCR and qPCR with SYBR green were sequenced bidirectionally using the same primers and BigDye version 3.1 (Applied Biosystems, Foster City, CA, USA) according to the manufacturer’s instructions. After purification, the products were sequenced on an ABI Prism 3130 Genetic Analyzer (Applied Biosystems). The sequences obtained were then compared with those of other flaviviruses, alphaviruses, and leptospiral sequences using the BLASTn program (available online: blast.ncbi.nlm.nih.gov/Blast.cgi; accessed on 8 October 2021).

### 4.7. Statistical Analysis

The statistical analysis was performed with IBM SPSS program version 23. Clinical and biological characteristics were compared using a Chi^2^ test. Student’s *t*-test was used for normally distributed quantitative variables. Descriptive statistics were performed for the laboratory results. Variables with a *p*-value < 0.05 were considered significant.

## 5. Conclusions

Ultimately, diagnosis was achieved in 94% of cases using different diagnostic techniques. This study is the first to identify the presence of ZIKV at the beginning of 2015. This study is also the first to report the presence of *L. kmetyi* in the southern border region of Mexico.

## Figures and Tables

**Figure 1 pathogens-10-01450-f001:**
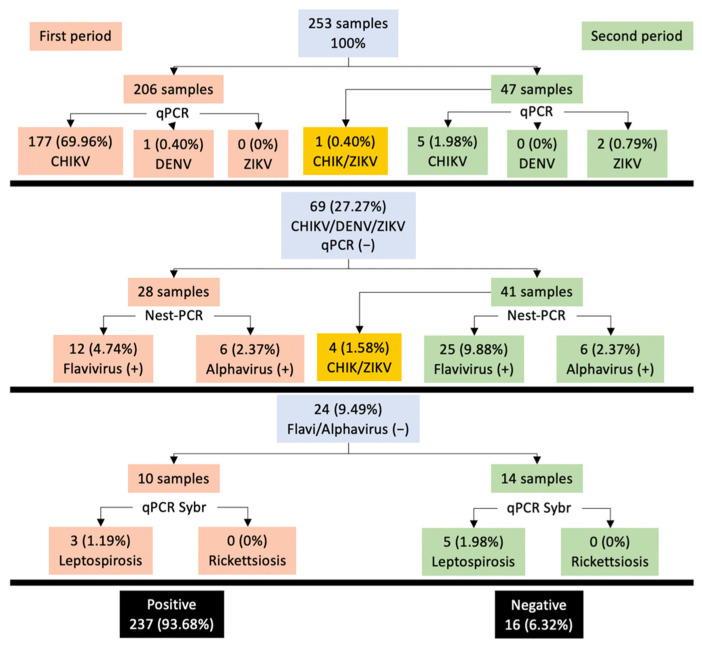
Epidemiological data were obtained for arboviruses and other pathogens. Gray boxes represent the total samples analyzed in each segment: (A) CHIKV/DENV/ZIKV, qPCR; (B) Flavi/Alphavirus, nested PCR; (C) Leptospira/Rickettsia, qPCR SybrGreen; orange and green boxes represent the samples collected in the two different periods (April–June 2015, and February–March 2016); the black boxes are the total positive and negative samples identified.

**Figure 2 pathogens-10-01450-f002:**
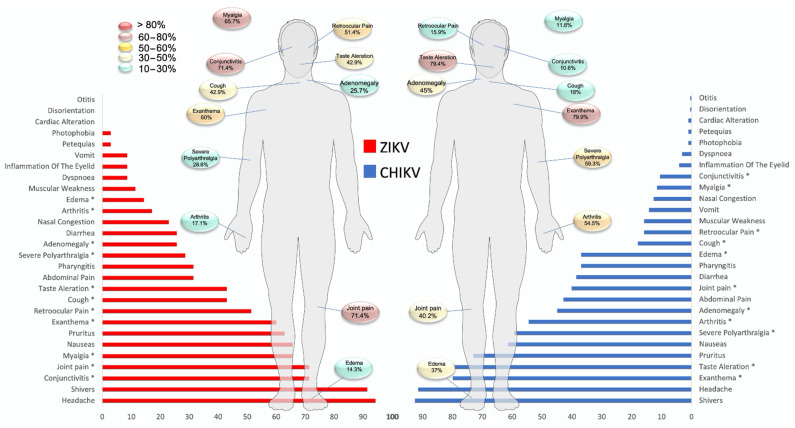
Comparison of clinical variables between the groups, CHIKV and ZIKV. ZIKV, Zika virus; CHIKV, chikungunya virus. ^a^ ZIKV+ vs. CHIKV+. * *p* < 0.05. Fisher’s exact test.

**Table 1 pathogens-10-01450-t001:** Primers/probes used in this study.

Primer and Probe	Sequence (5′–3′)	Target	Reference
**Probe**			
ZIKV 1086	CCGCTGCCCAACACAAG	E	[43]
ZIKV 1162c	CCACTAACGTTCTTTTGCAGACAT	E	[43]
ZIKV 1107-FAM	FAM-AGCCTACCTTGACAAGCAGTCAGACACTCAA-TAMRA	E	[43]
CHIK856	ACCATCGGTGTTCCATCTAAAG	nsP1	[3]
CHIK962c	GCCTGGGCTCATCGTTATT	nsP1	[3]
CHIK908-FAM	FAM-ACAGTGGTTTCGTGTGAGGGCTAC-TAMRA	nsP1	[3]
DENV 10635	GARAGACCAGAGATCCTGCTGTCT	3′UTR	[44]
DENV 10682	ACCATTCCATTTTCTGGCGTT	3′UTR	[44]
DENV123-10663-FAM	FAM-AGCATCATTCCAGGCAC-MGB	3′UTR	[44]
DENV4-10663-FAM	FAM-AACATCAATCCAGGCAC-MGB	3′UTR	This study
**Nested PCR**			
cFD2	GTGTCCCAGCCGGCGGTGTCATCAGC	ns5	[45]
MAMD	AACATGATGGGRAARAGRGARAA	ns5	[45]
FS 778	AARGGHAGYMCDGCHATHTGGT	ns5	[45]
ALPHA-1-Fod	TTTAAGTTTGGTGCGATGATGAAGTC	nsP4	[46]
ALPHA-1-Rev	GCATCTATGATATTGACTTCCATGTT	nsP4	[46]
ALPHA-2-Fod	GGTGCGATGATGAAGTCTGGGATGT	nsP4	[46]
ALPHA-2-Rev	CTATGATATTGACTTCCATGTTCAKCCA	nsP4	[46]
**Sybr-Green**			
RICK-ADN-For	TATGCTTGCGGCTGTCGGTTCTC	gltA	[47]
RICK-ADN-Rev	TTGCGGTAAGTTCGTAGTCTGCTTCTT	gltA	[47]
LEP-ADN-For	AGCAGCCGCGGTAATACGTATGG	16S rRNA	[47]
LEP-ADN-Rev	TTTAGGGCGTGGATTACTGGGG	16S rRNA	[47]

## Data Availability

Data are available upon request to the authors.

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
