# Peer review of "Molecular Characterization of Associated Pathogens in Febrile Patients during Inter-Epidemic Periods of Urban Arboviral Diseases in Tapachula Southern Mexico"

_pathogens, 2021, doi:10.3390/pathogens10111450_

Round 1

Reviewer 1 Report

Major Comments:

Line 82: In Figure 1: The description of Flavi/Alphavirus, nested PCR methods should added more information that the Nest-PCR products were further sequenced and blast as ZIKV/CHIKV sequence. 

Minor Comments:

  1. Line 70: “ and 88 (32.2%) were male,”  should be “ and 88 (33.2%) were male,”
  2. Line 76: “ and 5.93%(n=15) negative results”  should be “ and 6.32%(n=16) negative results”
  3. Line 88: “ most of the patients (six)”  should be “ most of the patients (6/8)”
  4. Line 98: “ Only 28% ZIKV-positive patients” – How 28% value is calculated from?
  5. Line 99: “ In patients with CHIKV, 16.9%” – How 16.9% value is calculated from?
  6. Line 100: “ For Group II, out of 51 patients” – What is the Group II means?
  7. Line 119: “ detection rate was 88.83%.” — How 88.83% value is calculate from?
  8. Line120: “ 57.44% prevalence” –How 57.44% value is calculate from? 

Author Response

Question.    Line 82: In Figure 1: The description of Flavi/Alphavirus, nested PCR methods should added more information that the Nest-PCR  products were further sequenced and blast as ZIKV/CHIKV sequence. 

Answer.    In the line 270 in the section of methodology was added.

Question.    Line 70: “ and 88 (32.2%) were male,”  should be “ and 88 (33.2%) were male,”

Answer. this change was made

Question.    Line 76: “ and 5.93%(n=15) negative results”  should be “ and 6.32%(n=16) negative results”

Answer. this change was made

Question.    Line 88: “ most of the patients (six)”  should be “ most of the patients (6/8)”

Answer. this change was made

Question.    Line 98: “ Only 28% ZIKV-positive patients” – How 28% value is calculated from?

Answer. References for standard clinical features are found on line 178-179

The most common clinical symptoms of chikungunya virus infection are acute fever and polyarthralgia, en nuestro studio 10/35 cumplian con este criterio. 

Therefore added in the text  28.5% (10/35) – line 97

Question.    Line 99: “ In patients with CHIKV, 16.9%” – How 16.9% value is calculated from?

Answer. References for standard clinical features are found on line 178-179

The most common clinical symptoms of Zika  virus infection are Fever, rash, arthritis and/or arthralgia and/or myalgia, conjunctivitis, and fatigue, en nuestro studio 32 (32/189) cumplian con este criterio.

Therefore added in the text 16.9% (32/189) – line 99

Question. Line 100: “ For Group II, out of 51 patients” – What is the Group II means?

Answer. this number was corrected to 47 patients, since they correspond to the group of February-March 2016

Question. Line 119: “ detection rate was 88.83%.” — How 88.83% value is calculate from?

Answer. The calculation was carried out with data from the period April - June 2015, to be able to compare with studies carried out on similar dates.
It is added on the line 124 88.83% (183/206).

Question. Line120: “ 57.44% prevalence” –How 57.44% value is calculate from? 

Answer. The calculation was carried out with data from the February-March 2016 period, in order to be able to compare with studies carried out on similar dates.

It is added on the line 125 in 57.44% (183/206).

Reviewer 2 Report

The manuscript by Calvo-Anguiano et al. reports data on molecular characterization of associated pathogens in febrile patients during inter-epidemics periods of urban arboviral diseases, in Tapachula southern Mexico.

Overall, the research is well described and the paper intend to be the first documented survey on ZIKV circulation in human population since the beginning of 2015 and the first report on L. kmetyi in the southern border region of Mexico.  

The results of the study are worth to be published after revision and some changes that have to be performed (list below).

Line 26: please change “or” with “and”

Line 76: please change “june” with “June”

Line 77: you say that there were eight patients with leptospirosis, but in Table S1 are 7

Line 133: maybe you mean emergence, because emergency means something else / I think is better to use the syntagma “emergence and reemergence” all over the article

Line 156: please change the capital letter "K" with lower case "k" in “L. Kmetyi”

Table S1: please add “the” in front of 253

Figure 1, Table S1 and table S2: please check once again the numbers because in table S2 there are 59 male positive for CHIKV and in table S1 are 59 (April-June 2015) + 1 (February-March 2016)/ also in S1 there are 7 female identified ZIKV positive between April-June 2015 and 14 between February-March 2016 and in S2 there is a total of 20 positive female samples, and also for Leptospirosis 5 female positive samples in S1 versus 6 samples in S2.

Maybe it will be better to verify once again all the numbers.

You do not mention anything about vector population presented in the collection sites. What is the situation relating vector competence and vector transmission in these regions? Are there studies, because if not you should mention this.

Did you check the travel situation of the subjects taken into the study?

In my opinion these points would need to be clarified prior to publication.

Author Response

Question. Line 26: please change “or” with “and”

Answer. This change was made.

Question. Line 76: please change “june” with “June”

Answer. This change was made.

Question. Line 77: you say that there were eight patients with leptospirosis, but in Table S1 are 7

Answer. This change was made en supplementary material.

Question. Line 133: maybe you mean emergence, because emergency means something else / I think is better to use the syntagma “emergence and reemergence” all over the article.

Answer. This change was made in line 140.

Question. Line 156: please change the capital letter "K" with lower case "k" in “L. Kmetyi”

Answer. This change was made in line 164

Question. Table S1: please add “the” in front of 253

Answer. This change was made

Question. Figure 1, Table S1 and table S2: please check once again the numbers because in table S2 there are 59 male positive for CHIKV and in table S1 are 59 (April-June 2015) + 1 (February-March 2016)/also in S1 there are 7 female identified ZIKV positive between April-June 2015 and 14 between February-March 2016 and in S2 there is a total of 20 positive female samples, and also for Leptospirosis 5 female positive samples in S1 versus 6 samples in S2.

Answer. This change was made

Question. You do not mention anything about vector population presented in the collection sites. What is the situation relating vector competence and vector transmission in these regions? Are there studies, because if not you should mention this.

Answer. This change was made. The following text was added (lines 113-118).

An important factor associated with these diseases is the presence and abundance of vector mosquitos; numerous Aedes aegypti populations are found in the region, and vector incrimination was demonstrated after the isolation of CHIKV and ZIKV from field mosquito collections [6, 17, 18]. The vertical transmission of DENV was also documented by Danis-Lozano et al. (2019) in an extensive egg survey from south Mexico states [19].

the changes suggested for the English language have already been made by a language expert.